# Organophosphate Pesticides and Pyrethroids in Farmland of the Pearl River Delta, China: Regional Residue, Distributions and Risks

**DOI:** 10.3390/ijerph20021017

**Published:** 2023-01-05

**Authors:** Runlin Yao, Siyu Yao, Tao Ai, Jiahui Huang, Yang Liu, Jianteng Sun

**Affiliations:** 1Guangdong Provincial Key Laboratory of Petrochemical Pollution Processes and Control, School of Environmental Science and Engineering, Guangdong University of Petrochemical Technology, Maoming 525000, China; 2Bathurst Future Agri-Tech Institute, Qingdao Agricultural University, Qingdao 266109, China; 3Department of Environmental Sciences, College of Earth and Environment Sciences, Lanzhou University, Lanzhou 730000, China

**Keywords:** organophosphate pesticides, pyrethroids, distribution characteristic, vertical profiles, risk assessment

## Abstract

A regional-scale survey was conducted to assess the occurrence, distribution, and risk of two extensively used pesticides (organophosphate pesticides and pyrethroids) in agricultural soils from the Pearl River Delta (PRD), South China. All target organophosphate pesticides (OPPs) and pyrethroids (PYs) were detected in the soil samples and both with a detection rate of 100%. The residues of the sum of six OPPs and the sum of four PYs were in the range of LOD–991 ng/g and 8.76–2810 ng/g, respectively. Dimethoate was the dominant OPPs, and fenpropathrin was the predominant PYs in the soils of the PRD region. With intensive agricultural activities, higher residues of OPPs and PYs in soils were detected closer to the seaside, among which Zhuhai city and Huizhou city suffered more serious combined pesticide pollution. The vertical compositional profiles showed that dimethoate could be detected through each soil layer in the PRD region’s nine cities. The human exposure estimation of OPPs showed insignificant risks to the local population. In contrast, cypermethrin and fenpropathrin showed a potential ecological risk of 2.5% and 3.75% of the sampling sites, respectively. These results can facilitate those commonly used pesticide controls and promote sustainable soil management.

## 1. Introduction

Pesticide application is the most effective pest control regime to boost agricultural product yield [1,2]. However, only approximately 0.1% of applied pesticides can effectively act against the target organisms, and large amounts of those spraying pesticides will eventually accumulate in the soils [3,4]. Those pesticides and their derivatives residues in the soil will destroy soil biodiversity, enter into the human body via the food chain, and finally pose a potential threat to human health [5,6]. Given their frequent detection in agricultural soils and their potential adverse effects on ecosystems and human health, the occurrence and distribution of pesticides have attracted widespread interest on a global scale [7,8]. Therefore, evaluating the levels of commonly used pesticides and their potential human and ecological risk in agricultural soils is of great practical significance.

Among the various agricultural pesticides, organophosphorus pesticides (OPPs) are one of the most powerful pesticides used to control and exterminate pests [9]. However, OPPs were found to accumulate in the environment for long periods owing to their frequent application and moderate persistence [10,11]. Residues of OPPs have been frequently detected in agricultural soils [12,13]. For example, the concentrations of OPPs in agricultural soils ranged from 1.23 to 239 ng/g, with an average concentration of 23 ng/g, of which chlorpyrifos was the main compound found in most soil samples from Nepal [4]. It has been confirmed that those OPPs, such as dimethoate and parathion, can affect the central nervous systems and inhibit the decomposition of the human neurotransmitter acetylcholine [14,15]. Moreover, a case study to describe exposure to pesticides in general populations indicated that OPPs were found ubiquitously in urine samples from eight countries in rapidly developing countries in Asia [16]. Therefore, human health risks caused by OPPs contamination in agricultural soils have attracted increasing concern worldwide [17,18]. Therefore, there is a need to investigate the status of OPPs pollution in soils to evaluate the possible human exposure risks at a regional scale.

Pyrethroids (PYs) could effectively control numerous pests with high selective toxicity, and they are less toxic to non-target organisms than organochlorine pesticides (OCPs) and OPPs [19]. Due to their good traits, PYs are extensively used in agricultural crops nowadays, leading to their frequent detection in crops and soils worldwide [20]. Those PYs residues in the soils would have adverse impacts on soil biology and change the activity of enzymes, ultimately affecting plant growth and soil fertility [21]. It was demonstrated that PYs were toxic to earthworms in the soil [22]. Moreover, PYs can accumulate in many kinds of vegetables and fruits grown in contaminated soil [20], which further increases the risk of PYs exposure to human beings. Moreover, it is claimed that PYs can harm mammals’ central nervous system, reproductivity, and immune system at high exposure doses [23]. However, few studies focused on the distribution of PYs in soils and their potential risk assessment on a regional scale.

Pearl River Delta (PRD), located in the central and southern parts of Guangdong Province in South China, has long been a highly developed agricultural zone. The part of PRD consists of Dongguan (DG), Jiangmen (JM), Zhongshan (ZS), Huizhou (HZ), Guangzhou (GZ), Shenzhen (SZ), Zhaoqing (ZQ), Zhuhai (ZH) and Foshan (FS). Along with the rapid urbanization of Guangdong Province, the urban cultivated areas have shrunk significantly. Various insecticides are increasingly being used to increase agricultural production to meet the food needs of population growth. Six neonicotinoid insecticides were detected in the soils of an agricultural zone in Zengcheng within the PRD [24]. Legacy and current-use insecticides in soils of the PRD and surrounding areas were examined, with a mean concentration of ∑7DDX and ∑4HCH levels of 18.4 ng/g and 14.2 ng/g, respectively [25]. With the heavy use of pesticides in modern agricultural production, elevated concentrations of OPPs and PYs often coexist in agricultural soils. However, studies that specifically address the distribution and characteristics of commonly used OPPs and PYs in agricultural soils in the PRD region are still lacking.

In recent years, the demand for OPPs was decreased while that for PYs was increased. In 2022, the national demand for OPPs was about 7.19 million tons, a decrease of 6.29% compared with the previous year. While the demand for PYs was about 0.38 million tons in 2022, an increase of 8.4% over the previous year [26]. Thus, an extensive survey was conducted in the PRD area to reveal the pattern of the combined OPPs and PYs contamination in agricultural soils. The main objectives of this work are (i) to investigate the residual concentrations and spatial distributions of OPPs and PYs in agricultural soils of the PRD region; (ii) to characterize the vertical distribution of pesticides in soil profiles; and (iii) to assess the ecological and health risk of OPPs and PYs contamination in soils. These findings have important implications for risk assessment and management of OPPs and PYs pollution in agricultural soils.

## 2. Materials and Methods

### 2.1. Chemicals and Reagents

In the chemical analysis, six standards of OPPs were dichlorvous, systos, dimethoate, malathion, malathion, and methyl parathion. Meanwhile, the four standards of PYs were cypermethrin, fenpropathion, cyhalothoin, and deltamethrin. The stock standards were purchased from AccuStandard (New Haven, CT, USA). We used analytical standards of ≥98% purity. Acetone and hexane were pesticide grade, obtained from Thermo Fisher Scientific (Pittsburgh, PA, USA). All other chemicals and reagents used in this study were of analytical reagent grade or higher purity.

### 2.2. Sample Collection

A total of 240 surface soil samples (0–15 cm) were collected from agricultural regions in the Pearl River Delta (PRD) from June to September 2019, of which 18 sampling sites were located in Dongguan (DG), 30 were in Jiangmen (JM), 16 were in Zhongshan (ZS), 28 were in Huizhou (HZ), 48 were in Guangzhou (GZ), 30 were in Shenzhen (SZ), 24 were in Zhaoqing (ZQ), 19 were in Zhuhai (ZH), and 27 were in Foshan (FS). The distribution of sampling points is shown in Appendix A. In addition, a total of 72 soil samples from different depths with 10 cm intervals were also collected. The soil types mainly included yellow sandy, black sandy, and yellow powder. A detailed procedure for soil sampling can be referred to in our previous study [21]. Briefly, five subsamples were collected with a bamboo scoop at each site and mixed into a single sample. The collected soil samples were immediately transported to a laboratory at a low temperature and stored at −20 °C for subsequent analysis.

### 2.3. Sample Extraction, Cleanup, and Analysis

All samples were freeze-dried, ground, and sieved through 75 mesh stainless steel screens. For OPPs, a 5.0 g soil sample was extracted with 60 mL acetone/petroleum ether (4:1, *v*/*v*) in an ultrasonic bath three times. The extract was then concentrated with the use of a rotary evaporator, purified through a florisil column, and recovered in 60 mL hexane/acetone (4:1, *v*/*v*). The elution was concentrated and exchanged into 1 mL hexane for GC-MS analysis. Details about the analysis process were referred to in previous works [10,27].

The extraction and analysis of PYs were adapted following the reported procedure [21,28]. Briefly, an aliquot of 5 g soil sample was ultrasonically extracted three times with 15 mL hexane. The supernatant obtained by centrifugation was then concentrated in a rotary evaporator and further purified through an alumina: silica gel (1:1, *w*/*w*) column. The extracts were eluted with 50 mL of dichloromethane: hexane (1:1, *v*/*v*) and concentrated to 1 mL of hexane for subsequent GC-MS analysis. The detailed analysis procedure of PYs was provided in previous work [21].

### 2.4. Quality Assurance and Quality Control

A blank and a sample duplicate were analyzed with each batch of fifteen soil samples. Procedural blanks were used to exclude contamination and interference in the whole procedure. The levels of target pesticides detected in the laboratory blanks were below the limit of quantification. The recovery rates of six OPPs and four PYs in the spiked samples ranged from 78.4% to 103.6% and 71.5% to 93.2%, respectively. The variations in duplicate samples were lower than 15% (*n* = 3). Relative standard deviations were <10%.

### 2.5. Health and Ecological Risk Evaluation

On the basis of the OPPs concentrations in the agricultural soils from the PRD region, we assessed the exposure to the non-cancer risk of OPPs using the health index (HI) method recommended by the US EPA [29]. In this study, soil ingestion and dermal contact are the two main pathways for human exposure to soil OPPs. The following formulas were used to calculate the HI values:(1)$$\mathrm{ADDingest}=\frac{\mathrm{Csoil}\text{×}\mathrm{IRS}\text{×}\mathrm{CF}\text{×}\mathrm{EF}\text{×}\mathrm{ED}}{\mathrm{BW}\text{×}\mathrm{AT}}$$
(2)$$\mathrm{ADDdermal}=\frac{\mathrm{Csoil}\text{×}\mathrm{CF}\text{×}\mathrm{SA}\text{×}\mathrm{AF}\text{×}\mathrm{ABS}\text{×}\mathrm{EF}\text{×}\mathrm{ED}}{\mathrm{BW}\text{×}\mathrm{AT}}$$
(3)$$\mathrm{HQ}=\frac{\mathrm{ADD}}{\mathrm{RfD}}$$
(4)$$\mathrm{HI}=\sum {\mathrm{HQ}}_{i}=\sum \frac{{\mathrm{ADD}}_{i}}{{\mathrm{RfD}}_{i}}$$
where HQ is the hazard quotient and ADD is the average daily dose (mg kg^−1^ day^−1^). If a HI value is below 1, then there is no potential health risk to local inhabitants [10]. All human health risk assessment parameters and their reference value are listed in the supporting information.

On the basis of the PYs concentrations in the agricultural soils from the PRD region, we assessed the ecological risk using the risk quotient (RQ) method. Potential ecological risks posed by PYs were expressed as the ratio of the soil PYs residues detected in this study to the lethal dose (LC_50_) of terrestrial organisms (e.g., earthworms) [30,31]. The following formulas were used to calculate the RQ values:(5)$$\mathrm{RQ}=\mathrm{RQi}=\sum \frac{{\mathrm{MEC}}_{i}}{{\mathrm{PNEC}}_{i}}$$
(6)$$\mathrm{PNEC}=\frac{{\min(\mathrm{LC}}_{50})}{\mathrm{AF}}$$
where MEC is the measured concentration of PYs in agricultural soils (mg·kg^−1^), and PNEC is representative of the predicted no-effect concentration, which is equivalent to the Screening Benchmark given by USEPA. PNECs for earthworms were obtained by dividing the 14-day LC_50_ value by an AF of 1000. i represents the different categories of PYs. If an RQ value is less than 1, then there is no potential ecological risk. If RQ ≥ 1, then there is a potential ecological risk [32]. The LC_50_ value of earthworms comes from the database of PPDB (Pesticide Properties DataBase) and PMEP (Pesticide Management Education Program). The links to the database of PPDB and PMEP are http://sitem.herts.ac.uk/aeru/ppdb/en/index.htm (accessed on 1 January 2023) and https://pmep.cce.cornell.edu/ (accessed on 1 January 2023), respectively.

### 2.6. Statistical Analysis and Spatial Mapping

Statistical differences in this study were determined by one-way ANOVA with the least significant difference post-hoc test (SPSS version 22.0). Statistical significance was considered as *p* < 0.05. Spearman correlation analysis was conducted using SPSS 22.0. The geostatistical interpolation method known as ‘Kriging’ (spherical model) was used to examine how OPPs and PYs were spatially distributed in ArcGIS 10.2 (ESRI, Redlands, CA, USA).

## 3. Results and Discussion

### 3.1. Occurrence of Pesticides in Soils in PRD

Table 1 summarizes the concentrations (ng/g dry weight) of target OPPs and PYs in topsoil samples collected from the PRD region. The total residual concentrations of OPPs ranged from <LOD to 991 ng/g, with an average concentration of 145 ng/g and a detection rate of 100%, indicating ubiquitous pollution by OPPs in agricultural soils of the PRD region. Dimethoate was measured at the highest average concentration (49.7 ng/g) with the highest detection rate of 97.1%. Similarly, dimethoate was also found to be the most dominant OPPs in soils from the Yangtze River Delta of China, with a mean concentration of 50.8 ng/g and the highest detection frequency of 80.9% [10]. Moreover, dimethoate was also found in orange and nuts samples across China [33,34]. Methyl parathion was ubiquitously present (96.7% detection frequency) compared to parathion (21.7%), with a mean concentration of 47.7 ng/g. Dichlorvos (52.1%), demeton (50.8%), and malathion (63.3%) were also frequently detected, but their mean concentrations were all below 18.2 ng/g. Among those detected OPPs, only malathion is approved for use in the EU at present [2]. Parathion and methyl parathion, those two OPPs have been withdrawn or banned in many countries, are still detected in the soils of this region. Therefore, much more attention needs to be paid to those OPPs residues in soils in this studied area.

The detection rates of PYs in the tested soil samples reached 100%, indicating their wide distribution in the PRD region. Total concentrations of the four PYs ranged from 8.76 ng/g to 2810 ng/g with a mean value of 109 ng/g, and the order of average concentration of the four individual PYs was as follows: fenpropathrin (85.3 ng/g) > cyhalothrin (11.6 ng/g) > cypermethrin (10.1 ng/g) > deltamethrin (1.52 ng/g). This compositional profile was consistent with a previous finding in agricultural soils collected from the Yangtze River Delta of China but was different from the patterns of PYs in soil collected from PRD with cypermethrin as the dominant component [21,25]. Specifically, fenpropathion was predominant and was the most commonly detected PYs, with a detection rate of up to 100%. Direct pesticide applications to crops are presumably the dominant source of fenpropathrin residues found in the studied area. Moreover, a relatively long half-life of fenpropathrin in the soil might have slowed its degradation [35], which helps explain its high residual concentrations in soils. Cypermethrin was the least detected PYs, with a detection rate of only 37.5% in our study, which was contrary to cypermethrin was the most frequently detected PYs in China because of its low price [20]. A possible reason for this phenomenon was that fenpropathrin was the preferred insecticide rather than cypermethrin in the PRD region. Deltamethrin and cyhalothrin were detected with the highest concentration at 32.0 ng/g and 739 ng/g, with detection frequencies of 68.8% and 99.6%, respectively. Cyhalothrin residues were also detected in honey, earthworm, and urine samples in populations in eight countries [16,36,37]. Moreover, PYs can transport into various vegetables and fruits to further increase their human exposure risk [38]. It is thus imperative to investigate the characteristics and risks of PYs in agricultural soils at a regional scale.

Multiple residues found in one soil sample were mainly caused by applying different pesticides (OPPs and PYs) [39]. Obviously, the soil samples containing four, five, and multiple residues were noted (Figure 1), suggesting that the soil has suffered from combined pollution caused by multiple pesticides. The residues of pesticides found in the highest detectable rate were 7, with a detectable rate of 27.1%. Six and eight pesticides were detected at a similar level, at rates of 22.1% and 24.2%, respectively. In total, 10 kinds of pesticide residues were found in one soil sample. Both OPPs and PYs are classified into insecticides. Altering the spray of those two insecticides to reduce pesticide resistance would make their coexistence in the soils [40,41]. However, the coexistence of various pesticides in agricultural soils can cause synergistic effects that lead to more serious environmental and human health implications, even for low pesticide concentrations [42,43]. Therefore, paying more attention to the combined pollution of multiple pesticides in agricultural soils is vitally important.

The correlation analysis between the six OPPs and four PYs in agricultural soils of the PRD region is shown in Figure 2. It was obvious that demeton strongly correlates with malathion, dimethoate, methyl parathion, and fenpropathrin (*p* < 0.001). Based on the correlation analysis, the analyses can speculate some characteristics of those coexisting pesticides. Firstly, those pesticides are likely to originate from the same source or can be classified into the same kind of pollutants. OPPs and PYs are both classified into insecticides, and they are recommended to be applied alternately or together to lower the risk of exposure [44]. Secondly, several compounds would probably show similar adsorption behavior following a period of prolonged pollution, although they belong to a different class of pollutants. The predominant PYs of fenpropathrin were found to have a strong correlation with several OPPs, including demeton, dimethoate, and methyl parathion, but with no significant correlation with those other PYs (*p* < 0.001). These differences could be caused by the different physicochemical properties of these pesticides.

### 3.2. Spatial Distribution of Pesticides

The soil residual levels of OPPs in different cities of the PRD region are shown in Appendix A. It is obvious that the residual levels of the six OPPs clearly varied among the different cities (Appendix A). The mean values of ∑6OPPs in the nine cities of the PRD region were as follows: ZS (198 ng/g) > ZH (176 ng/g) > HZ (157 ng/g) > SZ (154 ng/g) > FS (142 ng/g) > GZ (138 ng/g) > JM (123 ng/g) > DG (121 ng/g) > ZQ (117 ng/g). Obviously, the mean value of six OPPs in ZH was significantly higher than the city of ZQ, DG, and JM (*p* < 0.05). The spatial distribution of total OPPs in the agricultural soils across the PRD region is shown in Figure 3A. The soils from the ZS, ZH, and HZ had significantly high OPPs levels, while those soils from ZQ and DG had low OPPs residues in the soil. This could be due to the intense use of pesticides for agriculture in those cities. Moreover, wastewater irrigation may be another important potential source of OPPs [45]. Therefore, another plausible explanation for this spatial pattern was that those soils closer to the seaside receive wastewater irrigation from highly urbanized districts, such as ZS, ZH, and HZ. Previous studies have shown that OPPs were widely detected in water bodies [1]. It was reported that the main soil type in Guangdong Province was red soils, usually with slightly acid soils. Acidic soils were more easily subjected to OPPs pollution since those soils with low pH values would limit the hydrolysis of OPPs [46,47].

It is observed that the concentrations of PYs vary significantly across cities in the PRD region (Appendix A). Significant higher total concentrations of PYs were observed in the city of HZ, ZS, DG, and JM in the PRD region (*p* < 0.05). Similarly, PYs concentrations in soils from HZ, especially in the northeast and southeast, were significantly higher than in soils from ZQ and ZH (Figure 3B). Possible reasons for the spatial distribution differences of PYs are the amount and frequency of pesticide applications [28]. In addition, it was found that land use types have a great influence on the levels of PYs residues in soil. Our previous study indicated that those soils near Taihu Lake had suffered more serious PYs contamination in Jiangsu and Zhejiang provinces, where agricultural activities are more developed in these regions [21].

High residues of OPPs and PYs were both found in the soils closer to the seaside, among which ZH and HZ suffered more serious combined pesticide pollution. However, OPPs and PYs were both found to be at a low level in agricultural soils from ZQ. The humid and warm weather in Guangdong province facilitates the propagation of pests [24]. OPPs and PYs were classified as insecticides and are widely used for pest control in Guangdong province as the main sources of pesticide contamination in the soil environment are intensive agricultural activities [10]. The differences in the spatial distribution of OPPs and PYs could be caused by the different pesticide use habits of local villagers. Moreover, the concentration of OPPs in agricultural soils was higher than those PYs in most of the sampled areas except for the southeastern part of GZ. This result indicated that farmers tend to use OPPs rather than PYs in the PRD region.

### 3.3. Vertical Distribution of Pesticides in Soil Profiles

Figure 4 and Appendix A depict the vertical distribution of OPPs and PYs in the representative soil samples collected from different depths in the study site. In general, the pesticide concentration varied depth by depth. Dimethoate could be detected across the entire soil profile in all nine cities of the PRD region, indicating their intensive application and their good leaching potential. Its high polar structure and relatively weak sorption onto soil particles (K_OC_ ranges from 16.25 to 51.88 L/kg) make it prone to leaching and are proven with high leaching risk [48,49]. Indeed, dimethoate was frequently detected in groundwater on a global scale [50,51,52]. In ZH, higher concentrations of dichlorvos and demeton were observed in the 0–20 cm top soils but quickly declined to a very low level when the soil depth was >20 cm. However, a homogeneous distribution of dichlorvos was detected throughout the whole soil profile in JM and ZS. Parathion in the entire soil profile of DG was almost all below the detection limit.

For PYs, the dominant fenpropathrin was distributed homogeneously along the soil profile in DG, SZ, and ZS. The higher concentration of fenpropathrin was found in the 30–40 cm and 70–80 cm of the two soil layers in ZQ, and the highest concentration was distributed in the deepest 80 cm soil profile in HZ. A higher concentration of fenpropathrin was detected in the 20–40 cm layer of the soil profile, but with a dramatic decrease in the 50–90 cm soil layer in JM. Cyhalothrin showed a relatively uniform distribution trend in ZS, SZ, HZ, ZH, and DG. However, it was only detected in the 10–40 cm soil layer in JM. Similarly, cyhalothrin was only detected in the 30–50 cm soil layers in ZQ and JM. In addition, cypermethrin and deltamethrin were almost not detected in the soil layers, except for a few points that were detected with low concentrations.

### 3.4. Ecological and Health Risk Assessment

The non-cancer risk posed by exposure to OPPs through ingestion and dermal contact with the soil is extensively compared in Figure 5. The HI values of the total OPPs were 0.00035 and 0.00326 for adults and children, indicating that non-cancer risk is negligible in soils of the PRD region. The average non-cancer risk of OPPs was 9 times greater for children than adults, among which the risk of ingestion for children was 10 times higher than for adults, and the risk of dermal contact in children was 6 times higher than that for adults. Consistent with most previous studies, the non-cancer risks to children were higher than those to adults in soils [53]. Dimethoate showed the highest mean value of non-cancer risk (HI = 0.00112 and 0.00012 for children and adults), followed by methyl parathion (HI = 0.00108 and 0.000116 for children and adults), and finally, demeton (HI = 0.000410 and 0.000044 for children and adults).

The potential ecological risk posed by PYs to the target terrestrial organisms (earthworms) was evaluated using the RQ method. The soil RQ results for each sampling point are shown in Figure 5. Few sPRDs (a total of 18 sampling sites) had excessive levels of total ecological risk, with an exceeding rate of 7.5% in this studied area. Individual PYs residues with ecological risk (RQ ≥ 1) include cypermethrin and fenpropathrin, with an exceeding rate of 2.5% and 3.75% in all the 240 detected soil samples, respectively. For cypermethrin, a total of six sampling points were found with ecological risk, among which FS and DG each with two points, while SZ and GZ each with one point. For fenpropathrin, there are nine sampling points with ecological risk, among which GZ with three points, ZS and DG with two points, and HZ and JM each with one point. These results indicated that cypermethrin and fenpropathrin applied in those cities might pose risks to the ecological environment. It was especially noted that the RQ value was as high as 15.3 in the southeastern of HZ. The other two PYs of cyhalothrin and deltamethrin were found to have low potential ecological risks. However, some of the PYs residues in the agricultural soils may be transferred to the crops, posing a potential threat to human health. Therefore, more attention should be paid to the potential ecological risk and health risks of PYs in further studies.

## 4. Conclusions

This study investigated the occurrence, distribution, and potential risks of target OPPs and PYs in agricultural soils from the PRD region. The six OPPs and four PYs were detected in all topsoil samples, with a mean concentration of 145 ng/g and 109 ng/g, respectively. Dimethoate was the most dominant OPPs compound, while fenpropathrin was the predominant PYs. Different OPPs and PYs concentration distributions in different regions were presented in the PRD region. Moreover, dimethoate was easier to infiltrate into the soil, resulting in potential risks to the deep soil layers and groundwater. Among the target PYs, cypermethrin and fenpropathrin were found to pose a potential ecological risk to terrestrial organisms. The human health risks caused by OPPs pollution in soil were generally negligible. This research was valuable for understanding the pollution status, spatial distribution, and risk of OPPs and PYs in agricultural soils from the PRD region.

## Figures and Tables

**Figure 1 ijerph-20-01017-f001:**
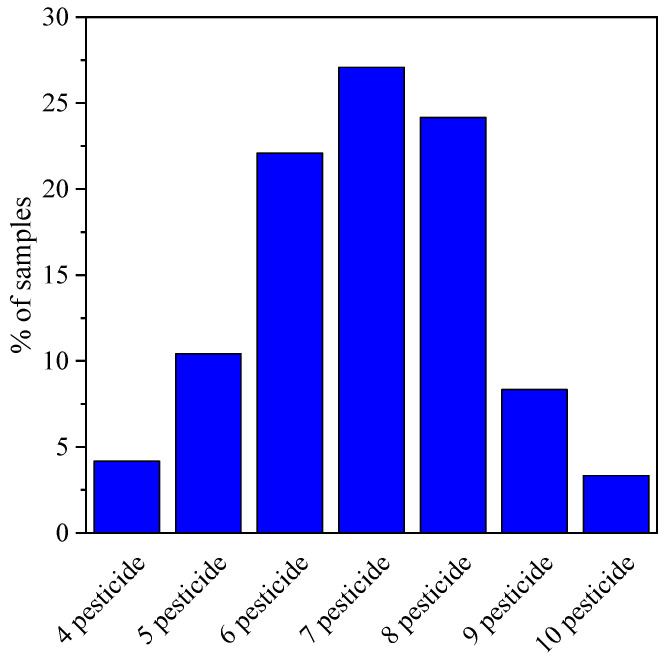
The multiple pesticide residues in the soil samples.

**Figure 2 ijerph-20-01017-f002:**
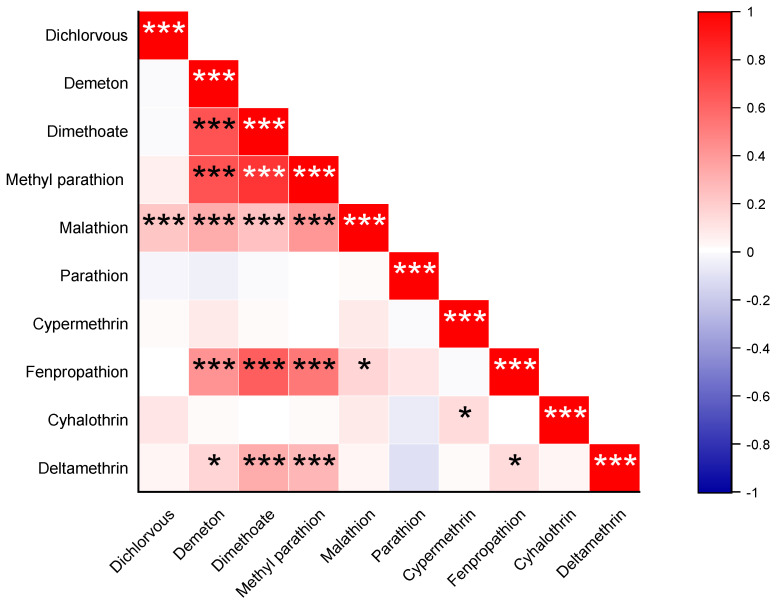
Correlation analysis between the 6 OPPs and 4 PYs in agricultural soils of the PRD region. * Indicates significance level at 0.05. *** Indicates significance levels at 0.001.

**Figure 3 ijerph-20-01017-f003:**
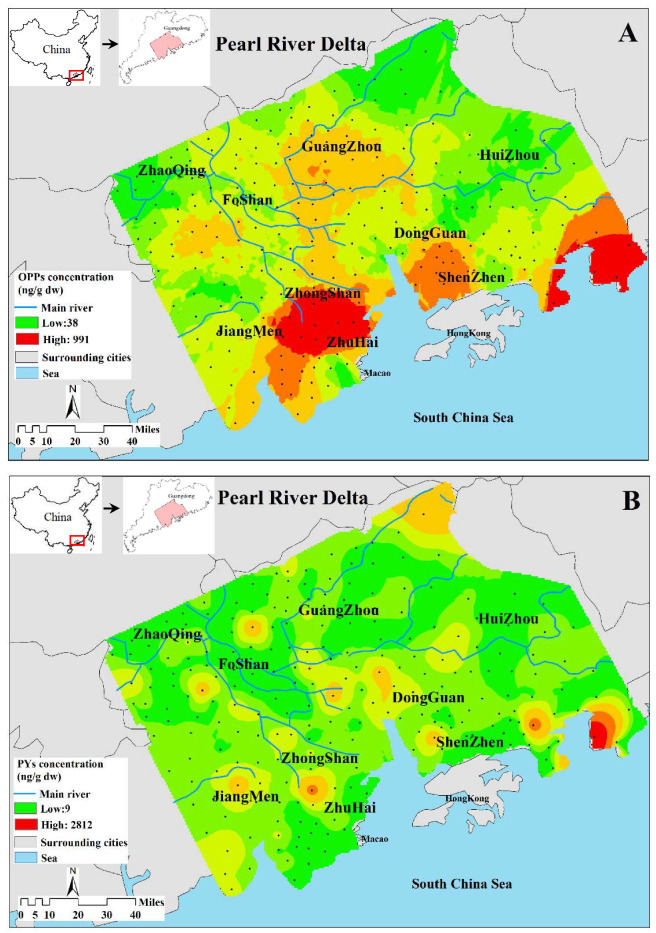
The spatial distributions of OPPs (**A**) and PYs (**B**) in agricultural soils of the PRD region.

**Figure 4 ijerph-20-01017-f004:**
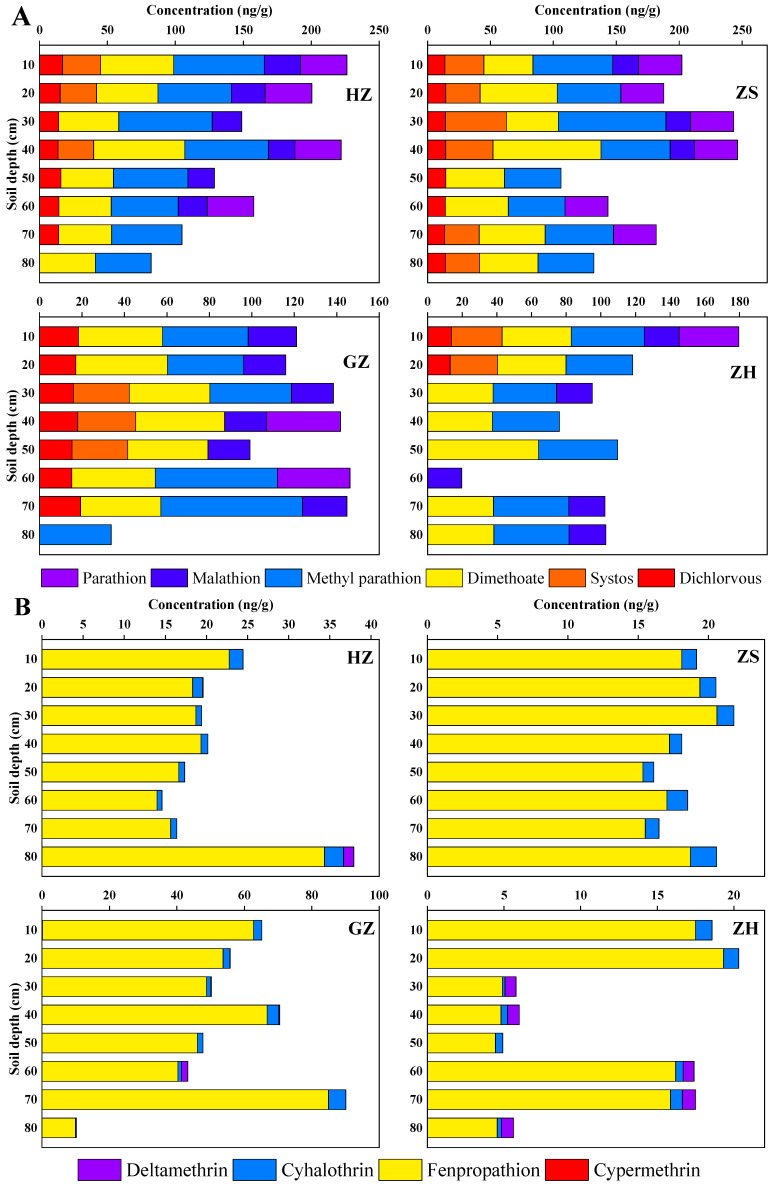
Vertical distributions of the concentrations and compositions of OPPs (**A**) and PYs (**B**) in agricultural soils of the PRD region.

**Figure 5 ijerph-20-01017-f005:**
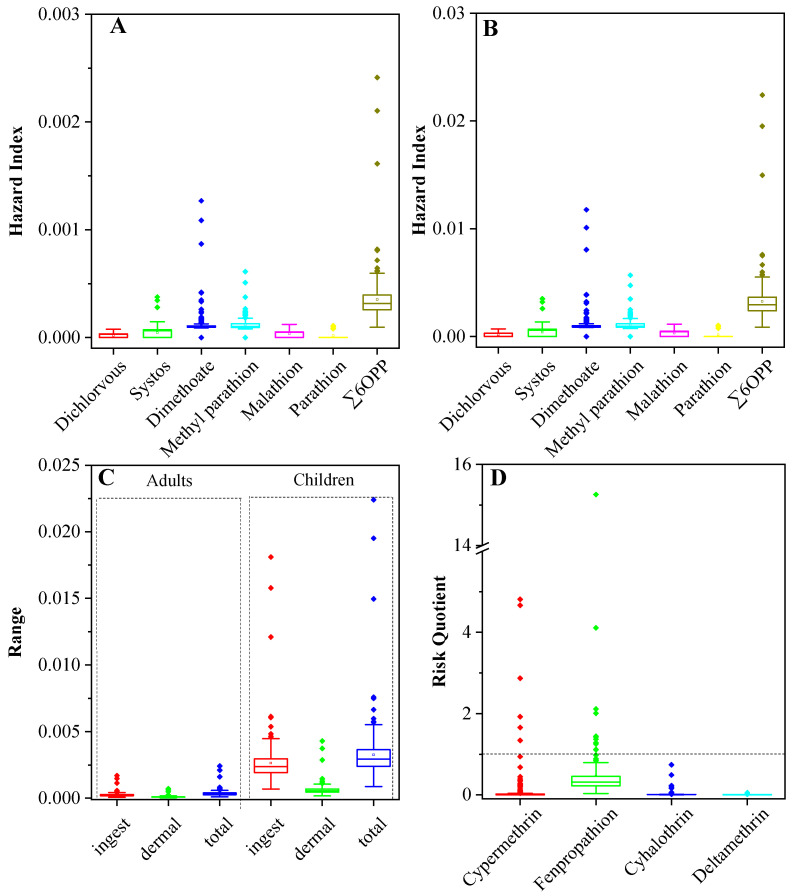
Non-cancer risks of OPPs to adults (**A**) and children (**B**), comparison of the contributions of different exposure pathways (**C**), and ecological risk of PYs in agricultural soils of the PRD region (**D**).

**Table 1 ijerph-20-01017-t001:** The concentration of the target OPPs and PYs in soils from the PRD region.

Compound	Detection Rate (%)	Mean (ng/g)	Min. (ng/g)	Max. (ng/g)	Median (ng/g)	CV (%)
Dichlorvos	52.1	7.34	ND	31.0	13.6	98.9
Demeton	50.8	18.2	ND	155	29.2	121
Dimethoate	97.1	49.7	ND	521	40.3	98.1
Malathion	63.3	14.1	ND	50.4	21.1	78.4
Parathion	21.7	7.52	ND	44.8	34.3	50.7
Methyl parathion	96.7	47.7	ND	251	21.1	191
Σ6OPPs	100	145	38.4	991	130	63.8
Cypermethrin	37.5	10.1	ND	481	2.51	20.0
Fenpropathion	100	85.3	5.25	2800	57.9	45.1
Cyhalothoin	99.6	11.6	ND	739	2.96	19.3
Deltamethrin	68.8	1.52	ND	32.0	0.579	45.2
Σ4PYs	100	109	8.76	2810	65.3	52.4

ND: not detected. CV: coefficient of variation. Σ6OPPs: the sum of the detected six OPPs. Σ4PYs: the sum of the detected six PYs.

## Data Availability

Not applicable.

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
