# Peer review of "Organophosphate Pesticides and Pyrethroids in Farmland of the Pearl River Delta, China: Regional Residue, Distributions and Risks"

_ijerph, 2023, doi:10.3390/ijerph20021017_

Round 1
Reviewer 1 Report
In the manuscript titled “Organophosphate Pesticide and Pyrethroids in Farmland of the Pearl River Delta, China: Regional Residue, Distributions and Risks”, the Authors investigated the residual concentrations and spatial distributions of OPPs (Organophosphorus Pesticides) and PYs (Pyrethroids) in agricultural soils of the Pearl River Delta region, characterizing vertical distribution of pesticides in soil profiles.
The objectives of the research are appropriate for INTERNATIONAL JOURNAL ENVIRONMENTAL RESEARCH AND PUBLIC HEALTH (IF: 4.54). This manuscript is well written in English, the contents are well argued and supported by an adequate number of bibliographical references. Moreover, this manuscript allows to study the pollution level of soil caused by the extensively use of these kind of contaminants and to assess the respective ecological and health risk, since literature lacks of this information.
Therefore, my final opinion is to accept the paper with minor revision.
Minor points:
1) In the Materials and Methods section, in the Paragraph 2.5. “Health and ecological risk evaluation”, at line 153, please remove the space between the round bracket and “and PMEP.”
2) In the Results and discussion section, in the Paragraph 3.1. “Occurrence of pesticides in soils in YRD” in the Figure 1 the words font is too small. Please, enlarge font.
3) In the Results and discussion section, in the Paragraph 3.1. “Occurrence of pesticides in soils in YRD” in the Figure 1 the number “4” is not clearly visible.
4) In the Results and discussion section, in the Paragraph 3.4. “Ecological and health risk assessment”, in the Figure 5 the words font in the graphic of comparison of the contributions of different exposure pathways (graphic C) is too small. Please, enlarge font.
Author Response
Response to Editors and Reviewers
Dear Bailey Xu
Associate Editor, International Journal of Environmental Research and Public Health
Thanks very much for the reviewers’ constructive comments on our manuscript entitled“Organophosphate Pesticides and Pyrethroids in Farmland of the Pearl River Delta, China: Regional Residue, Distributions and Risks” (Manuscript ID: ijerph- 2127637). We have attended all the comments carefully and made corrections (point by point) accordingly for which we hope to meet with the reviewers’ approval. The main revisions in the paper and the responses to the reviewers’ comments are presented below and in the response files.
Replies to Reviewer #1:
Comments: In the manuscript titled “Organophosphate Pesticide and Pyrethroids in Farmland of the Pearl River Delta, China: Regional Residue, Distributions and Risks”, the Authors investigated the residual concentrations and spatial distributions of OPPs (Organophosphorus Pesticides) and PYs (Pyrethroids) in agricultural soils of the Pearl River Delta region, characterizing vertical distribution of pesticides in soil profiles.
The objectives of the research are appropriate for INTERNATIONAL JOURNAL ENVIRONMENTAL RESEARCH AND PUBLIC HEALTH (IF: 4.54). This manuscript is well written in English, the contents are well argued and supported by an adequate number of bibliographical references. Moreover, this manuscript allows to study the pollution level of soil caused by the extensively use of these kind of contaminants and to assess the respective ecological and health risk, since literature lacks of this information.
Therefore, my final opinion is to accept the paper with minor revision.
Author Response:
The reviewer’s comments to our work are highly appreciated and thanks for the reviewer’s comments. We have revised the manuscript carefully to make it clearer than the previous version. The specific revisions are presented as follows.
1). In the Materials and Methods section, in the Paragraph 2.5. “Health and ecological risk evaluation”, at line 153, please remove the space between the round bracket and “and PMEP.”
Author Response:
Thanks for the reviewer’s careful inspection. We have removed the space between the round bracket and “and PMEP.” in our revised manuscript. (See Line 160)
2). In the Results and discussion section, in the Paragraph 3.1. “Occurrence of pesticides in soils in YRD” in the Figure 1 the words font is too small. Please, enlarge font.
Author Response:
Thanks for the reviewer’s comments. We have enlarged the words font of Figure 1 in the revised manuscript. (See Line 227)
3). In the Results and discussion section, in the Paragraph 3.1. “Occurrence of pesticides in soils in YRD” in the Figure 1 the number “4” is not clearly visible.
Author Response:
Thanks for the reviewer’s comments. We have enlarged the words font of Figure 1 in the revised manuscript, and we hope that this change will make these numbers clearly visible. Thanks for your good comments again. (See Line 227)
4). In the Results and discussion section, in the Paragraph 3.4. “Ecological and health risk assessment”, in the Figure 5 the words font in the graphic of comparison of the contributions of different exposure pathways (graphic C) is too small. Please, enlarge font.
Author Response:
Thanks for the reviewer’s comments. We are sorry we did not make it clear. We have enlarged the font of Figure 5 in the revised manuscript. (See Line 329)
Cordially
Jianteng Sun, Ph.D

Reviewer 2 Report
This is a very interesting article. However, I believe that the results and discussion section should be reviewed so it is clearer to the reader. Please consider my comments below:
Methods.
Authors should consider providing data or information available on the use of pesticides in the area monitored, quantities used and types/year, in order to justify their research.
Please also add the references to the USEPA methods that have been followed (subsection 2.5) in the manuscript as in the supplementary materials provided.
Provide the references/links to the database of PPDB (Pesticide Properties DataBase) and PMEP (Pesticide Management Education Program).
Results
Authors should justify the data provided in Table 1 for the pesticides that have been little detected in the samples such as parathion. For this specific pesticide, an average mean of 7.5 is provided, meanwhile the median is 0? How these data was statistically processed? Do they followed a normal or abnormal distribution?
Moreover, the legend in Table 1 should be strengthen to provide more information about the acronyms used.
Line 171 – the results presented from ref number 10, are they from the same area or another monitoring study? Please rewrite this sentence so it reads better. Same for the following sentence about the orange(s?) and nuts. Authors should describe these studies briefly, by including important information, such as the monitoring dates, and concentrations, so the reader has all the necessary information at a glance. This will be applicable to all the remaining section, e.g. line 200, which urine samples? This section should be carefully reviewed and strengthen.
Were pesticides only monitored in oranges?
Line 175 – describe what the concentration of 20 ng/g means.
Line 176 – is there any similar legislation in China?
Line 204 – this sentence is a bit confused, please rewrite providing more information.
Line 314 – why authors indicate that “dimethoate contributes substantial non-cancer risk soils of PRD region” if the risks identified are negligible?
